# Polymineralic Inclusions in Loparite-(Ce) from the Lovozero Alkaline Massif (Kola Peninsula, Russia): Hydrothermal Association in Miniature

Julia A. Mikhailova [1,2,*], Yakov A. Pakhomovsky [1], Ekaterina A. Selivanova [1,2] and Alena A. Kompanchenko [1]

1 Geological Institute, Kola Science Centre, Russian Academy of Sciences, 184209 Apatity, Russia; pakhom@geoksc.apatity.ru (Y.A.P.); selivanova@geoksc.apatity.ru (E.A.S.); a.kompanchenko@ksc.ru (A.A.K.)
2 Nanomaterials Research Centre, Kola Science Centre, Russian Academy of Sciences, 184209 Apatity, Russia
* Correspondence: j.mikhailova@ksc.ru; Tel.: +7-81555-79333

**Abstract:** Polymineralic inclusions in loparite-(Ce) in alkaline rocks from the Lovozero massif (Russia) were investigated using electron microprobe analysis, Raman spectroscopy, and X-ray diffraction. A total of 21 mineral species and two groups of minerals (pyrochlore- and labuntsovite-group minerals) were found in these inclusions. Minerals in loparite-hosted inclusions can be divided into two groups: (1) minerals found typically in rocks bearing loparite-(Ce) grains (groundmass minerals) such as aegirine, magnesio-arfvedsonite, potassic feldspar, albite, fluorapatite, etc.; and (2) minerals that were not found in the rock outside of the loparite-(Ce) grains. The latter include lorenzenite, labuntsovite-group minerals, minerals of the neptunite–manganoneptunite series, vinogradovite, catapleiite, fluorite, britholite-(Ce), barylite, genthelvite, and barite, found in the studied samples exclusively inside loparite-(Ce) crystals. The minerals of the second group are typical hydrothermal minerals. We assume that the skeletal crystals of loparite-(Ce), when growing, captured both co-crystallizing minerals and small drops of the mineral-forming solution. Such drops subsequently crystallized within the loparite-(Ce), resulting in the formation of a hydrothermal mineral association.

**Keywords:** Lovozero massif; loparite-(Ce); polymineralic inclusions; hydrothermal minerals

## 1. Introduction

Loparite-(Ce), ideally (Na,Ce,Sr)(Ce,Th)(Ti,Nb)$_2$O$_6$, is a ubiquitous accessory mineral of alkaline igneous rocks, including nepheline syenites and foidolites [1,2]. This mineral also occurs in carbonatites [3,4] and in contact with metamorphic and metasomatic rocks. Loparite was first discovered in nepheline syenites of the Lovozero peralkaline massif in 1890 by Wilhelm Ramsay, the first explorer of the geology of the Kola Peninsula, north-western Russia. Ramsay and Hackman [5] characterized this mineral as "mineral no. 1", and noted its similarity with perovskite, CaTiO$_3$. In 1925, I. G. Kuznetsov, using a sample of pegmatite from the neighboring Khibiny massif, described loparite as a new mineral species [6].

Loparite-(Ce) belongs to the perovskite subgroup of the eponymous supergroup of minerals that also includes perovskite, CaTiO$_3$, lueshite, NaNbO$_3$, tausonite, SrTiO$_3$, lakargiite, CaZrO$_3$, etc., [7]. The mineral occurs mostly as complex twins with a cubic and cubooctahedral habit. For a long time, the crystal structure of loparite-(Ce) had not been solved, as this mineral, in addition to exhibiting intricate twinning, usually contains inclusions of different minerals and shows a significant intragranular variation in composition. In 2000, Mitchell and colleagues [8] determined the symmetry of three loparite-(Ce) samples of differing composition using single-crystal X-ray diffraction data collected with a charge-coupled device area X-ray detector [9].

Loparite-(Ce) is a typical accessory mineral of various lithologies found in the Lovozero alkaline massif. In addition, some rocks of the massif are extremely enriched in loparite-(Ce), and these occurrences are exploited as a source of rare earth elements (REE), Ti, Ta, and

Nb. Currently, loparite ores are mined at the Karnasurt and Kedykvyrpakhk mines, located in the northwest of the massif. The study of the morphology and chemical composition of loparite-(Ce), as well as the genesis of loparite ores of the Lovozero massif, is the subject of many works [1,2,10–12]. The researchers suggested early magmatic or late magmatic [1,12], or hydrothermal origins [11,13,14] for loparite-(Ce).

In Lovozero's rocks, loparite-(Ce) usually forms fluorite-type twins of cubic and cubooctahedral habit saturated with polymineralic inclusions. Inclusions in minerals are widely used to elucidate the origin and evolution of mineral deposits. These small portions of the mineral-forming medium make it possible to identify the initial composition and reconstruct the physicochemical conditions. The study of inclusions in opaque minerals, such as minerals of the perovskite group, is a very difficult task. Nevertheless, studies of loparite-hosted inclusions in the Lovozero massif have been conducted. Indeed, previous studies have established that inclusions in loparite-(Ce) from the Karnasurt and Kedykvyrpakhk mines contain minerals typical of pegmatites and hydrothermal veins [11]. Such minerals include rhabdophane-(Ce), tsepinite-(Ca), barytolamprophyllite, rinkite, steenstrupine-(Ce), neptunite, manganoneptunite, and bornemanite.

In this work, we analyzed loparite-hosted inclusions from various rocks of the Lovozero massif. Using microprobe analysis, Raman spectroscopy, and X-ray diffraction analysis, we were able to significantly expand the list of minerals previously found in such inclusions. Based on the data obtained, we propose the mechanism for the formation of loparite-hosted inclusions and make assumptions about the conditions for the crystallization of loparite-(Ce).

## 2. Geological Background

The Lovozero peralkaline massif (Figure 1a), which formed around 360–370 Ma [15–17], is a layered laccolith located in the southwest of the Kola Peninsula (Russia) among Archean gneisses and granite-gneisses. The area of the Lovozero massif is 650 km$^2$. This is the second-largest alkaline pluton in the world after the Khibiny pluton. The massif consists of the following three major complexes [18–20]:

(1) The layered complex (Figure 1a) with a thickness of 1700 m occupies 77% of the massif's volume. This complex is called "layered" because it consists of a large number of subhorizontal layers (more commonly called "rhythms") of alkaline rocks (Figure 1b). The top of each rhythm consists of lujavrite. This is a trachytoid meso- or melanocratic nepheline syenite, consisting mainly of nepheline, microcline-perthite, aegirine-(augite), and alkaline amphiboles. Down the cross section of the rhythm, the content of mafic minerals gradually decreases, and lujavrite passes into massive leucocratic nepheline syenite, called foyaite. Toward the bottom of the rhythm, the content of feldspar gradually decreases, and foyaite passes into an almost monomineral nepheline rock called urtite. In some rhythms, urtite may be absent. Whereas transitions between rocks within rhythms are gradual, contacts between rhythms are sharp. Pegmatites are often located at the contact between the rhythms. All rhythms of the layered complex are grouped into seven series (I–VII from top to bottom). In each series, the urtite layers are additionally indicated by Arabian numerals. Figure 1a shows only some of the urtite horizons, namely I-4, II-4, II-7, III-1, III-10, III-14, IV-1, and IV-2.

(2) The eudialyte complex (18% of the massif's volume), with a thickness of 100 to 800 m, overlaps the layered complex. The eudialyte complex is not layered and consists mainly of lujavrite enriched with minerals of the eudialyte group. A small part of the eudialyte complex is foyaite, as well as fine-grained/porphyritic nepheline syenites, which usually form small lenses and sheet-like bodies.

(3) The rocks of the poikilitic complex form irregularly shaped bodies or lenses located among the rocks of both the eudialyte and the layered complexes (Figure 1a,c). The poikilitic complex (5% of the massif's volume) consists of poikilitic and uneven-grained feldspathoid syenites. A main feature of these rocks is the presence of large

(up to 10 cm in length) feldspar laths with numerous inclusions of feldspathoids (nepheline, sodalite, and vishnevite).

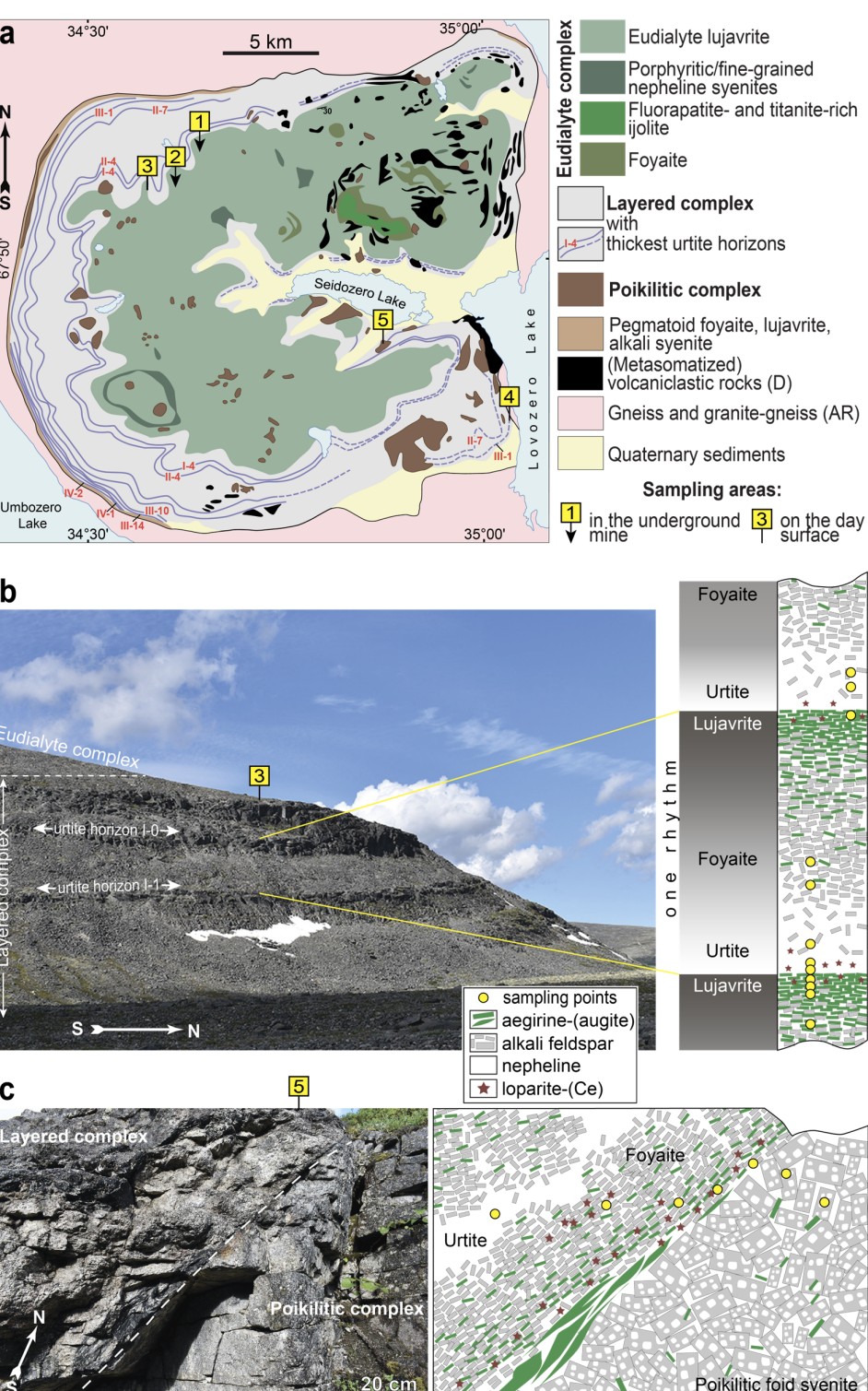

**Figure 1.** Geological background and sampling schemes: (**a**) geological scheme of the Lovozero alkaline massif after [21], with simplifications; (**b**) outcrop of the layered complex (left) and the schematic representation of an idealized rhythm [22] with sampling points (right). Lujavrite (tops of each rhythm) is more resistant to weathering, so individual rhythms are clearly visible in outcrops; and (**c**) outcrop of the contact of the layered and poikilitic complexes (left) and a scheme of this outcrop with sampling points (right). 1–5 are the numbers of sampling areas (see also Table 1).

Among the rocks of the layered and eudialyte complexes, there are a large number of roof xenoliths of the Devonian volcaniclastic rocks. The greatest number of large xenoliths occurs in the northeastern part of the massif. Here, volcaniclastic rocks have been preserved almost unchanged and represent an interbedding of olivine basalts, basalt tuffs, tuffites, and sandstones. In other parts of the massif, the xenoliths are smaller and intensely fenitized. Alkaline lamprophyre dikes were intruded at the final stage of the massif formation. Dikes are most common in the northwest and south of the massif and have a thickness of up to 6 m and a length of up to several hundred meters.

Loparite-(Ce) is a widespread mineral in the Lovozero massif, but the highest concentrations of this mineral are observed near contacts between different alkaline rocks. The contacts between overlying urtite and the underlying lujavrite in the layered complex (Figure 1b) are in places extremely (>10 vol. %) enriched with loparite-(Ce), forming the Lovozero loparite deposit [23]. In the layered complex, the Karnasurt and Kedykvyrpakhk mines exploit the loparite ores confined in the I-4 and II-4 urtite layers, and the Umbozero mine exploited the deeper III-10 and III-14 urtite layers. Local enrichment with loparite-(Ce) is also observed at the contacts between the layered and poikilitic complexes (Figure 1c).

## 3. Materials and Methods

For this study, 40 samples of loparite-bearing alkaline rocks from the Lovozero massif were sampled. Samples were taken along profiles crossing contacts of the rhythms of the layered complex (sampling areas 1–4 in Figure 1), as well as contacts of the layered and the poikilitic complexes (sampling area 5 in Figure 1). Table 1 shows a list of the studied samples and a brief description of their location.

**Table 1.** List of studied samples.

| Sampling Area (See Figure 1) | Location | Short Geological Description | List of Samples |
|---|---|---|---|
| *Contact zones between rhythms* | | | |
| 1 | Karnasurt underground mine (the Lovozero loparite deposit) | Narrow (0.1–0.4 m) loparite-rich horizon at the contact with the I-4 urtite layer and underlying lujavrite | LV-III-4-1, LV-III-4-2, LV-III-4-4, LV-III-5-2, LV-III-5-3, LV-III-5-4, LV-III-5-5, LV-III-6-1, LV-III-6-2, LV-III-6-5 |
| 2 | Kedykvyrpakhk underground mine (the Lovozero loparite deposit) | Narrow (0.1–0.4 m) loparite-rich horizon at the contact with the II-4 urtite and underlying lujavrite | LV-IV-3-5, LV-IV-3-1, LV-IV-3-2, LV-IV-1-2, LV-I-7, LV-I-8, LV-IV-1-1 |
| 3 | Outcrop on the slope of Mt. Kedykvyrpakhk 67°52′35.3″ N 34°34′37.7″ E | local loparite-rich areas at the contacts with the I-0 and I-1 urtite and underlying lujavrite | LV-316/1, LV-319/1, LV-319B, LV-319D, LV-333, LV-335/6, LV-335A, LV-335B, LV-335C, LV-335D, LV-336/1, LV-336/2 |
| 4 | Outcrop on the slope of Mt. Punkaruaiv 67°44′39.1″ N 35°01′44.6″ E | local loparite-rich areas at the contacts with III-1 urtite and underlying foyaite | LV-454, LV-454/2, LV-454/3 |
| *Contact zone between complexes* | | | |
| 5 | Outcrop on the slope of Mt. Ninchurt 67°47′27.7″ N 34°51′59.0″ E | local loparite-rich area at the contacts of the layered and poikilitic complexes | LV-484/2, LV-484/4, LV-484/5, LV-484/11, LV-484/12, LV-484/13 LV-487/1, LV-487/2 |

Back-scattered electron (BSE) images were obtained and minerals were diagnosed at the Geological Institute, Kola Science Center of the Russian Academy of Sciences (GI KSC RAS, Apatity, Russia), using the LEO-1450 scanning electron microscope (Carl Zeiss Microscopy, Oberkochen, Germany) with the energy-dispersive system (EDS) Aztec Ultimmax

100 (Oxford Instruments, Abingdon, UK) at 20 kV, 500–1000 pA, and with a 1–3 μm beam diameter. Electron microprobe analyses of minerals were performed at the GI KSC RAS using the Cameca MS-46 electron microprobe (Cameca, Gennevilliers, France) operating in the WDS-mode at 22 kV with a beam diameter of 5–10 μm, a beam current of 20–40 nA, and counting times of 10 s (for a peak) and 10 s (for background before and after the peak), with 5–10 counts for every element at each point. The following standards were used: lorenzenite (Na and Ti), pyrope (Al), wollastonite (Si and Ca), fluorapatite (P), $F_{10}S_{11}$ (Fe and S), atacamite (Cl), wadeite (K), metallic V, $MnCO_3$ (Mn), hematite (Fe), celestine (Sr), $ZrSiO_4$ (Zr), metallic Nb, baryte (Ba), $LaCeS_2$ (La and Ce), $LiPr(WO_4)_2$ (Pr), $LiNd(MoO_4)_2$ (Nd), $LiSm(MoO_4)_2$ (Sm), metallic Hf and Ta, thorite (Th), and metallic U. The analytical precision (reproducibility) of mineral analyses was 0.2–0.05 wt. % (2 standard deviations) for the major element and approximately 0.01 wt. % for impurities. The systematic errors were within the random errors.

The Raman spectra of minerals were recorded using a Horiba Jobin-Yvon LabRAM HR800 spectrometer equipped with an Olympus BX-41 microscope in the Saint-Petersburg State University (for labuntsovite-group minerals) and with an EnSpectr R532 (Spectr-M, ISSP RAS, Chernogolovka, Russia) spectrometer equipped with an Olympus BX-43 microscope in the Mining Institute KSC RAS (for other minerals). Raman spectra were excited using a solid-state laser (532 nm) with an actual power of 2 mW under the 50× objective (NA 0.75) (for labuntsovite-group minerals) and with an actual power of 18 mW under the 20× objective (NA 0.4) (for other minerals). The spectra were obtained in the range of 70–4000 $cm^{-1}$ at a resolution of 2 $cm^{-1}$ (for labuntsovite-group minerals) and of 5–8 $cm^{-1}$ (for other minerals) at room temperature. To improve the signal-to-noise ratio, the number of acquisitions was set to 15 (for labuntsovite-group minerals) and 20 (for other minerals). All spectra were processed using the algorithms implemented in the OriginPro 8.1 software package (Originlab Corporation, Northampton, MA, USA).

The X-ray diffraction (XRD) measurements were performed at the Kola Science Center of the Russian Academy of Sciences using a MiniFlex-600 powder diffractometer (Rigaku Corporation, Tokyo, Japan). The X-ray source was Cu Kα radiation. The tube current and the tube voltage were set at 15 mA and 40 kV, respectively. A one-dimensional detector (D/teX Ultra2, Rigaku Corporation, Tokyo, Japan) was used with a $K_\beta$ filter.

Mineral abbreviations (Table 2) are given in accordance with International Mineralogical Association (IMA)-approved mineral symbols [24], except for the eudialyte-, pyrochlore- and labuntsovite-group minerals.

**Table 2.** Abbreviations, names, and formulae of minerals mentioned in this article.

| Abbreviation | Mineral | Formula |
| --- | --- | --- |
| Ab | albite | $Na(AlSi_3O_8)$ |
| Aeg | aegirine | $NaFe^{3+}Si_2O_6$ |
| Anl | analcime | $Na(AlSi_2O_6) \cdot H_2O$ |
| Arf | arfvedsonite | $NaNa_2(Fe^{2+}_4Fe^{3+})Si_8O_{22}(OH)_2$ |
| Blmp | barytolamprophyllite | $(BaK)Ti_2Na_3Ti(Si_2O_7)_2O_2(OH)_2$ |
| Bri-Ce | britholite-(Ce) | $(Ce,Ca)_5(SiO_4)_3(OH)$ |
| By | barylite | $Be_2Ba(Si_2O_7)$ |
| Ctp | catapleiite | $Na_2Zr(Si_3O_9) \cdot 2H_2O$ |
| EGM | eudialyte-group mineral | $N_{15}M1_6M2_3M3M4Z_3[Si_{24}O_{73}]O'_4X_2$; N = Na, Ca, K, Sr, REE, Ba, Mn, $H_3O^+$; M1 = Ca, Mn, REE, Na, Sr, Fe; M2 = Fe, Mn, Na, Zr, Ta, Ti, K, Ba, $H_3O^+$; M3,4 = Si, Nb, Ti, W, Na; Z = Zr, Ti, Nb; O' = O, $OH^-$, $H_2O$; X = $H_2O$, Cl, F, $OH^-$, $CO_3^{2-}$, $SO_4^{2-}$ [25] |
| Fap | fluorapatite | $Ca_5(PO_4)_3F$ |
| Flr | fluorite | $CaF_2$ |
| Ghv | genthelvite | $Be_3Zn_4(SiO_4)_3S$ |
| Gon | gonnardite | $(Na,Ca)_2(Si,Al)_5O_{10} \cdot 3H_2O$ |
| Kfs | K-feldspar | $KAlSi_3O_8$ |

**Table 2.** *Cont.*

| Abbreviation | Mineral | Formula |
|---|---|---|
| Lmp | lamprophyllite | $(SrNa)Ti_2Na_3Ti(Si_2O_7)_2O_2(OH)_2$ |
| LGM | labuntsovite-supergroup mineral | [26] |
| Lom | lomonosovite | $Na_6Na_2Ti_2Na_2Ti_2(Si_2O_7)_2(PO_4)_2O_4$ |
| Lop-Ce | loparite-(Ce) | $(Na,Ce,Sr)(Ce,Th)(Ti,Nb)_2O_6$ |
| Lrz | lorenzenite | $Na_2Ti_2(Si_2O_6)O_3$ |
| Marf | magnesio-arfvedsonite | $NaNa_2(Mg_4Fe^{3+})Si_8O_{22}(OH)_2$ |
| Mnnpt | manganoneptunite | $KNa_2LiMn^{2+}_2Ti_2Si_8O_{24}$ |
| Nph | nepheline | $Na_3K(Al_4Si_4O_{16})$ |
| Npt | neptunite | $KNa_2LiFe^{2+}_2Ti_2Si_8O_{24}$ |
| Ntr | natrolite | $Na_2(Si_3Al_2)O_{10}\cdot2H_2O$ |
| Pcl | pyrochlore-group mineral | $A_{2-m}B_2X_{6-w}Y_{1-n}$; A = Na, Ca, Sr, Pb, Sn, Sb, Y, □; B = Ta, Nb, Ti, Sb, W; X = O; Y = □, $H_2O$, $OH^-$, O, F [27] |
| Rha-Ce | rhabdophane-(Ce) | $Ce(PO_4)\cdot H_2O$ |
| Sdl | sodalite | $Na_4(Si_3Al_3)O_{12}Cl$ |
| Vgd | vinogradovite | $Na_4Ti_4(Si_2O_6)_2[(Si,Al)_4O_{10}]O_4\cdot(H_2O,Na,K)_3$ |

□—vacancy.

## 4. Results

### 4.1. Loparite-(Ce) Morphology and Chemical Composition

In all samples studied, loparite-(Ce) is associated with rock-forming nepheline, microcline-perthite or orthoclase-perthite, aegirine, (magnesio-)arfvedsonite, and sodalite. The typical accessory minerals were fluorapatite, murmanite, lomonosovite, lovozerite- and eudialyte-group minerals. The samples differed mainly in the modal ratio of rock-forming minerals, rock texture (trachytoid in lujavrite, massive in foyaite and urtite, or poikilitic in rocks of the poikilitic complex), and intensity of secondary alteration. The main secondary minerals were natrolite, albite, and gonnardite. In addition, a typical secondary mineral is potassium feldspar, which forms small grains in close intergrowths with natrolite and/or albite (see Figure 2c). In the rocks of the Kedykvyrpakhk mine, secondary alterations are less intense compared to other sampling areas, and albite is the predominant secondary mineral here.

#### 4.1.1. Loparite-(Ce) Morphology in Contact Zones between Rhythms

In loparite ores of the Kedykvyrpakhk mine (Figure 2a), loparite-(Ce) usually forms well-faceted fluorite-type twins of cubic and cubooctahedral habit up to 2.5 mm across. Loparite-(Ce) from this mine rarely contains mineral inclusions.

Loparite-(Ce) from the Karnasurt mine (Figure 2b,c) usually forms crystals of cubic and cubooctahedral habit as well as rounded or irregularly shaped grains (up to 3 mm across). Usually, loparite-(Ce) contains numerous polymineralic inclusions, but grains without inclusions are also found (Figure 2c). In inclusion-bearing loparite-(Ce), the total volume of inclusions sometimes exceeds 50% of the host grain. Additionally, crystals and grains of loparite-(Ce) often contain bay-shaped cavities filled with groundmass minerals (i.e., minerals typical of the surrounding rock). For example, Figure 2d shows a grain of loparite-(Ce) containing numerous inclusions (shown by red arrows) and a bay-shaped cavity (circled by a dotted line). Note that loparite grains are always surrounded by secondary minerals, such as natrolite (Figure 2b,c), albite, and secondary K-feldspar (Figure 2c).

The morphology of loparite-(Ce) from local loparite-rich areas at the contacts between rhythms (sampling areas 3 and 4) is similar to that of loparite-(Ce) from the Karnasurt mine. Loparite-(Ce) crystals and grains containing both polymineralic inclusions and bay-shaped cavities filled with groundmass minerals are widespread here (Figure 2e,f). It should also be noted that loparite grains are always surrounded by natrolite (Figure 2e), as was observed in samples from the Karnasurt mine.

Rarely, anhedral loparite-(Ce) grains were found in the studied samples, filling the spaces between rock-forming minerals. Note that well-formed crystals of loparite-(Ce) and its anhedral grains are located in close proximity (Figure 2g). Moreover, some loparite-(Ce)

grains are intensively replaced by secondary minerals such as lamprophyllite, lomonosovite, and rhabdophane-(Ce) (Figure 2h).

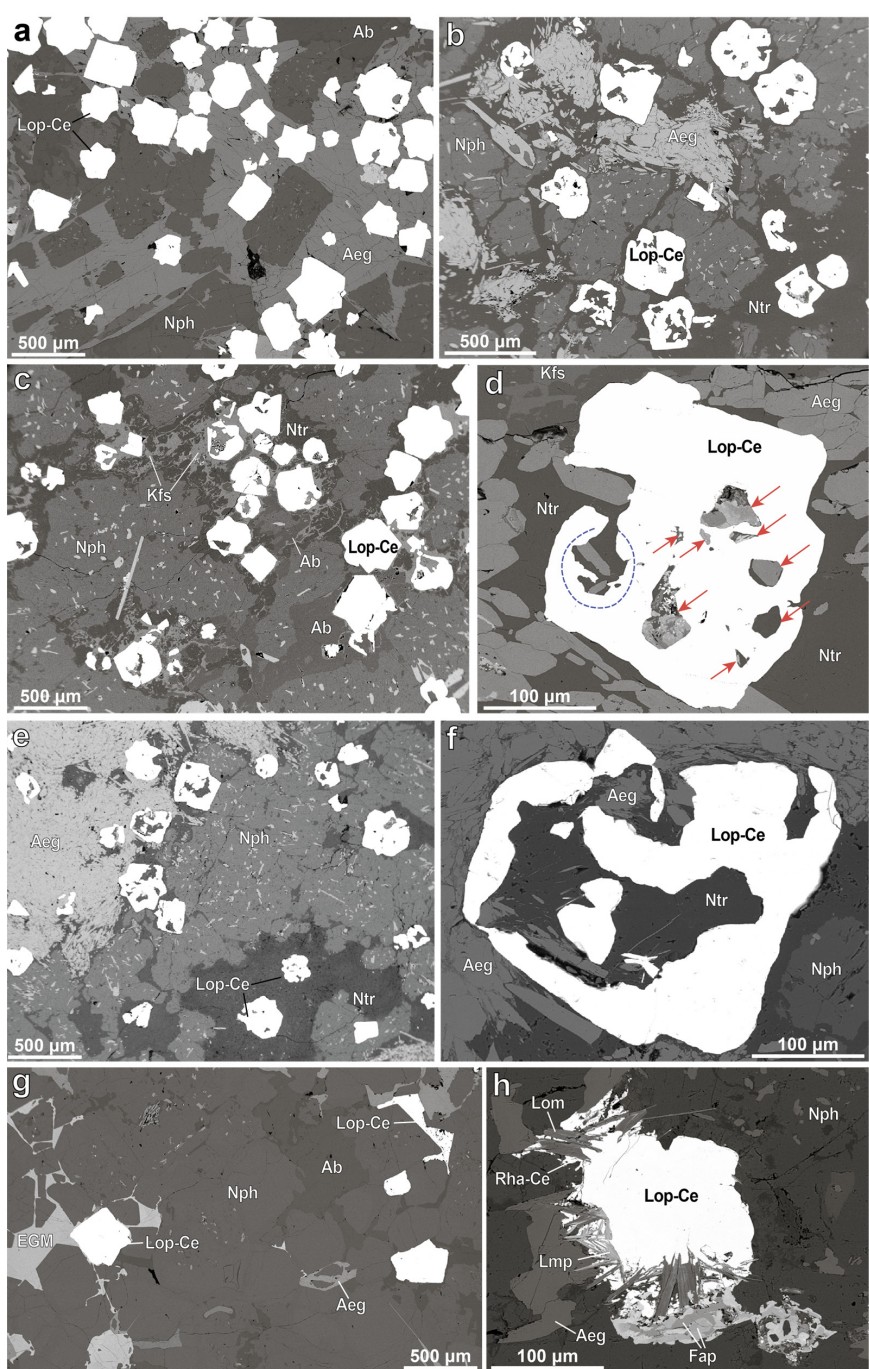

**Figure 2.** Loparite-(Ce) morphology in the contact zones between rhythms: (**a**) loparite-(Ce) crystals from urtite of the Kedykvyrpakhk mine. Sample LV-IV-3-1; (**b**,**c**) loparite-(Ce) crystals in urtite from the Karnasurt mine. Samples LV-III-6-1 (**b**) and LV-III-2-1 (**c**); (**d**) loparite-(Ce) crystals in urtite from the Karnasurt mine containing inclusions (shown by red arrows) and bay-shaped cavities (circled by a dotted line). Sample LV-III-6-1; (**e**) loparite-(Ce) crystals from urtite (sample LV-336/1) containing numerous inclusions and bay-shaped cavities; (**f**) loparite-(Ce) from urtite (sample LV-336/1) containing a bay-shaped cavity filled with groundmass minerals; (**g**) loparite-(Ce) crystals (bottom left and right) and its anhedral grain (top right) located in close proximity to each other. Sample LV-IV-1-2; and (**h**) replacement of loparite-(Ce) with secondary minerals. Sample LV-333. Back-scattered electron (BSE) images. Please see Table 2 for abbreviations.

### 4.1.2. Loparite-(Ce) Morphology in Contact Zone between Complexes

Enrichment with loparite-(Ce) is observed in a narrow zone at the contact between the rocks of the layered and poikilitic complexes, as shown in Figure 3a. Here, loparite-(Ce) forms crystals of cubooctahedral habit or rounded grains with numerous polymineralic inclusions (shown by red arrows in Figure 3b,c). In addition, most loparite-(Ce) crystals and grains contain bay-shaped cavities (circled by a dotted line, Figure 3b,c) filled with groundmass minerals. The average grain size of loparite-(Ce) is 0.6 mm across, and the maximum size is 4 mm across.

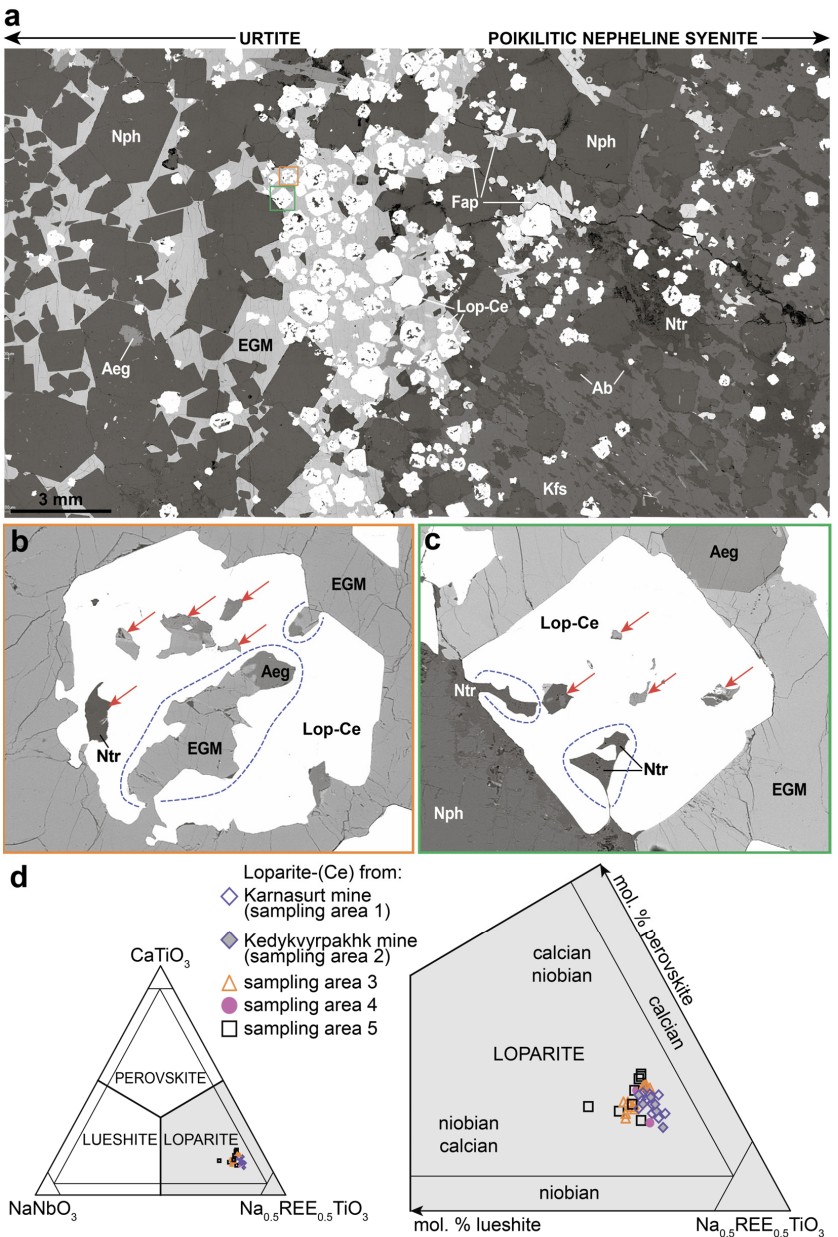

**Figure 3.** Loparite-(Ce) morphology in the contact zone between complexes (**a–c**) and the chemistry of loparite-(Ce) from all samples studied (**d**). (**a**) A loparite-rich zone at the contact of urtite (layered complex) and poikilitic nepheline syenite (poikilitic complex). Sample LV-484/13; (**b,c**) detailed images of Figure 3a (see the rectangles of the corresponding colors in Figure 3a). Loparite-(Ce) crystals contain inclusions (shown by red arrows) and bay-shaped cavities filled with groundmass minerals (circled by a dotted line). Back-scattered electron (BSE) images. See Table 2 for abbreviations; and (**d**) $CaTiO_3$ (perovskite)—$Na(REE)Ti_2O_6$ (loparite)—$NaNbO_3$ (lueshite) systematic of loparite-(Ce) [28].

### 4.1.3. Loparite-(Ce) Chemical Composition

Representative compositions of loparite-(Ce) are presented in Table 3. According to Mitchell and colleagues [7], the compositions of perovskite-group minerals can be expressed in terms of relatively few end-member compositions, namely $CaTiO_3$ (perovskite), $Na(REE)Ti_2O_6$ (loparite), $NaNbO_3$ (lueshite), $SrTiO_3$ (tausonite), $PbTiO_3$ (macedonite), $Ca_2Fe^{3+}NbO_6$ (latrappite), $Ca_2Nb_2O_7$, $REE_2Ti_2O_7$, $CaThO_3$, $CaZrO_3$, $KNbO_3$, and $BaTiO_3$.

**Table 3.** Representative compositions of loparite-(Ce).

| Sampling Area | 1 | 2 | 3 | 3 | 4 | 4 | 5 | 5 |
|---|---|---|---|---|---|---|---|---|
| Sample | LV-IV-3-5 | LV-III-6-2 | LV-335B | LV-336/1 | LV-454 | LV-454/3 | LV-484/13 | LV-484/11 |
| $Nb_2O_5$, wt. % | 6.92 | 6.99 | 6.72 | 7.20 | 8.40 | 7.88 | 7.02 | 8.86 |
| $Ta_2O_5$ | 0.70 | 0.63 | 0.57 | 0.20 | 0.80 | 0.63 | 0.66 | 0.82 |
| $TiO_2$ | 39.35 | 40.84 | 41.90 | 42.74 | 40.74 | 42.03 | 43.22 | 40.82 |
| $ThO_2$ | 0.27 | 0.47 | 0.85 | 0.93 | 0.65 | 0.84 | 0.77 | 0.82 |
| $Fe_2O_3$ | 0.32 | 0.42 | 0.24 | 0.21 | 0.40 | 0.19 | 0.19 | 0.22 |
| $La_2O_3$ | 9.25 | 9.15 | 8.90 | 8.76 | 9.29 | 8.54 | 8.32 | 8.98 |
| $Ce_2O_3$ | 17.52 | 17.45 | 18.07 | 17.68 | 19.42 | 18.11 | 17.46 | 18.37 |
| $Pr_2O_3$ | 1.73 | 1.52 | 1.14 | 1.33 | 1.21 | 0.83 | 0.94 | 1.19 |
| $Nd_2O_3$ | 4.04 | 4.05 | 4.47 | 4.51 | 3.76 | 3.77 | 3.85 | 3.78 |
| CaO | 4.59 | 4.81 | 5.38 | 5.47 | 3.93 | 5.24 | 5.61 | 3.93 |
| SrO | 3.65 | 3.95 | 3.31 | 2.93 | 1.78 | 3.05 | 3.71 | 3.30 |
| $Na_2O$ | 9.04 | 8.82 | 8.16 | 7.95 | 8.62 | 8.09 | 7.95 | 8.31 |
| $K_2O$ | bdl | bdl | 0.05 | bdl | 0.03 | 0.04 | 0.05 | 0.04 |
| Total | 97.35 | 99.08 | 99.76 | 99.90 | 99.03 | 99.22 | 99.75 | 99.44 |
| | | | | Formulae based on O = 3 pfu | | | | |
| Nb | 0.09 | 0.09 | 0.09 | 0.09 | 0.11 | 0.10 | 0.09 | 0.11 |
| Ta | 0.01 | - | - | - | 0.01 | - | 0.01 | 0.01 |
| Ti | 0.87 | 0.89 | 0.90 | 0.91 | 0.88 | 0.90 | 0.91 | 0.88 |
| Th | - | - | 0.01 | 0.01 | - | 0.01 | - | 0.01 |
| $Fe^{3+}$ | 0.01 | 0.01 | 0.01 | - | 0.01 | - | - | 0.01 |
| La | 0.10 | 0.10 | 0.09 | 0.09 | 0.10 | 0.09 | 0.09 | 0.10 |
| Ce | 0.19 | 0.18 | 0.19 | 0.18 | 0.20 | 0.19 | 0.18 | 0.19 |
| Pr | 0.02 | 0.02 | 0.01 | 0.01 | 0.01 | 0.01 | 0.01 | 0.01 |
| Nd | 0.04 | 0.04 | 0.05 | 0.05 | 0.04 | 0.04 | 0.04 | 0.04 |
| Ca | 0.14 | 0.15 | 0.16 | 0.17 | 0.12 | 0.16 | 0.17 | 0.12 |
| Sr | 0.06 | 0.07 | 0.05 | 0.05 | 0.03 | 0.05 | 0.06 | 0.05 |
| Na | 0.52 | 0.49 | 0.45 | 0.44 | 0.48 | 0.45 | 0.43 | 0.46 |
| | | | | Mol. % end members | | | | |
| loparite | 72 | 69 | 69 | 68 | 72 | 67 | 65 | 69 |
| perovskite | 11 | 14 | 17 | 17 | 12 | 16 | 18 | 12 |
| lueshite | 10 | 10 | 9 | 9 | 12 | 11 | 10 | 12 |
| tausonite | 6 | 7 | 6 | 5 | 3 | 5 | 6 | 6 |
| $ThTi_2O_6$ | - | - | - | 1 | 1 | 1 | 1 | 1 |

bdl—below detection limit; pfu—per formula unit.

Compositional data were recalculated into these end members using an Excel spreadsheet designed by Locock and Mitchell [28]. It has been established that most of the loparite grains studied belong to the $CaTiO_3$—$Na(REE)Ti_2O_6$—$NaNbO_3$ system (Figure 3d). They contain low levels of tausonite, $CaThO_3$, and other end members (Table 3). The composition of loparite-(Ce) varies slightly (Figure 3d). The lowest concentrations of lueshite and perovskite end members are found in loparite-(Ce) from the loparite mines (sampling areas 1 and 2). The highest concentrations of the above-mentioned end members are found in loparite-(Ce) from the contact zone between complexes (sampling area 5).

### 4.2. Inclusions in Loparite-(Ce)

As mentioned above, loparite-(Ce) contains numerous polymineralic inclusions unevenly distributed within the host mineral (Figures 2c–e, 3a–c and 4a,b). The shapes of

these inclusions are usually irregular, and the size varies from 5 to 500 μm across. The total volume occupied by inclusions can be up to half the volume of the loparite-(Ce) crystal. As a rule, each of the inclusions contains two to seven different minerals (for example, Figure 4). At the same time, the mineral associations of neighboring inclusions can differ significantly (Figure 4a,b). Figures 4–7 show typical mineral associations in inclusions found in loparite-(Ce) from the contact zones between rhythms (Figures 4 and 5) and from the contact zone between complexes (Figures 6 and 7).

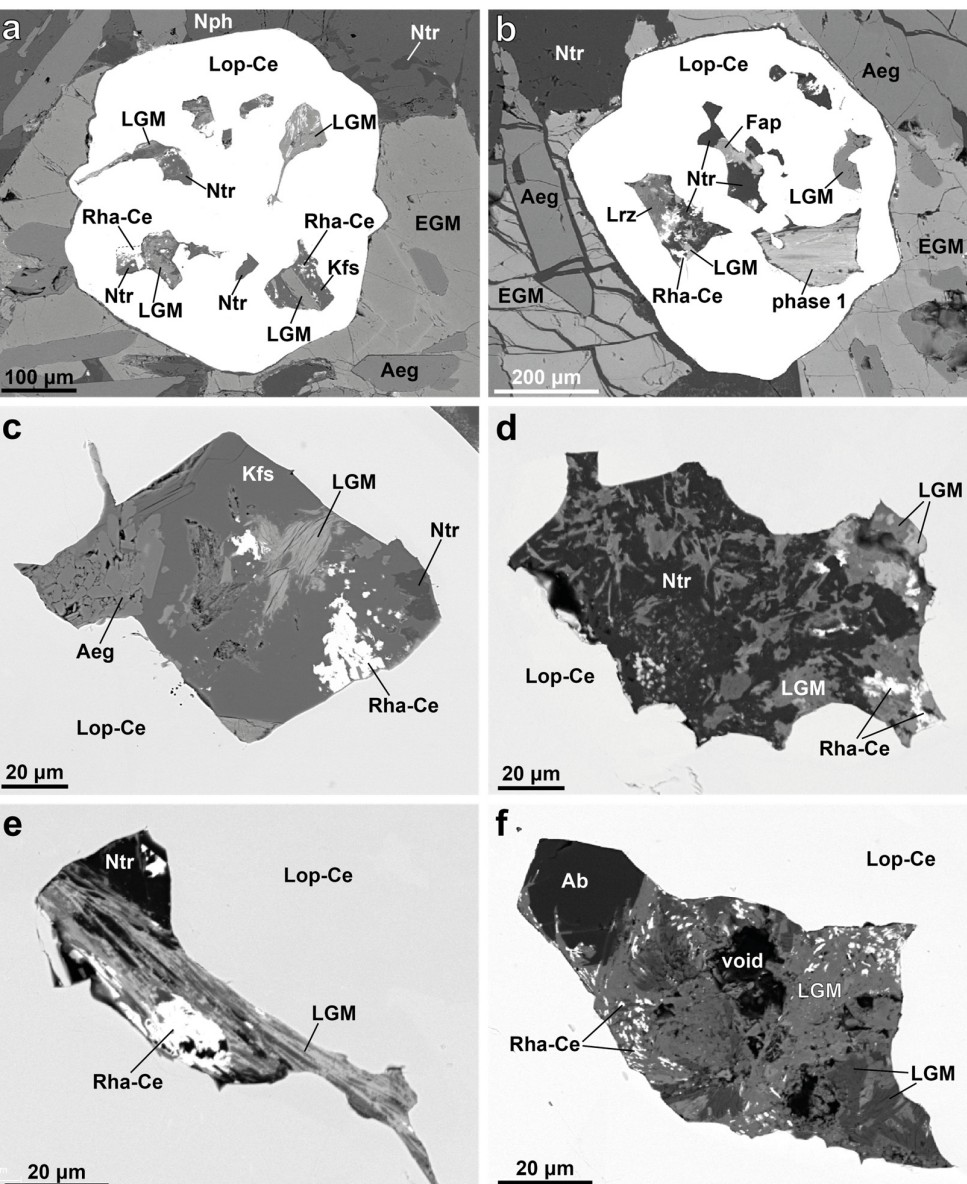

**Figure 4.** Back-scattered electron images of inclusions in loparite-(Ce) from the contact zones between rhythms: (**a**) mineral associations in inclusions from sample LV-319B; (**b**) mineral associations in inclusions from sample LV-336/2; (**c**) detailed image of one of the inclusions in loparite-(Ce) from sample LV-III-6-2; (**d**,**e**) mineral associations in inclusions from sample LV-336/1; and (**f**) mineral association in inclusions from sample LV-333. See Table 2 for abbreviations. phase 1 = undiagnosed Ca-(Ti,Nb)-Si phase.

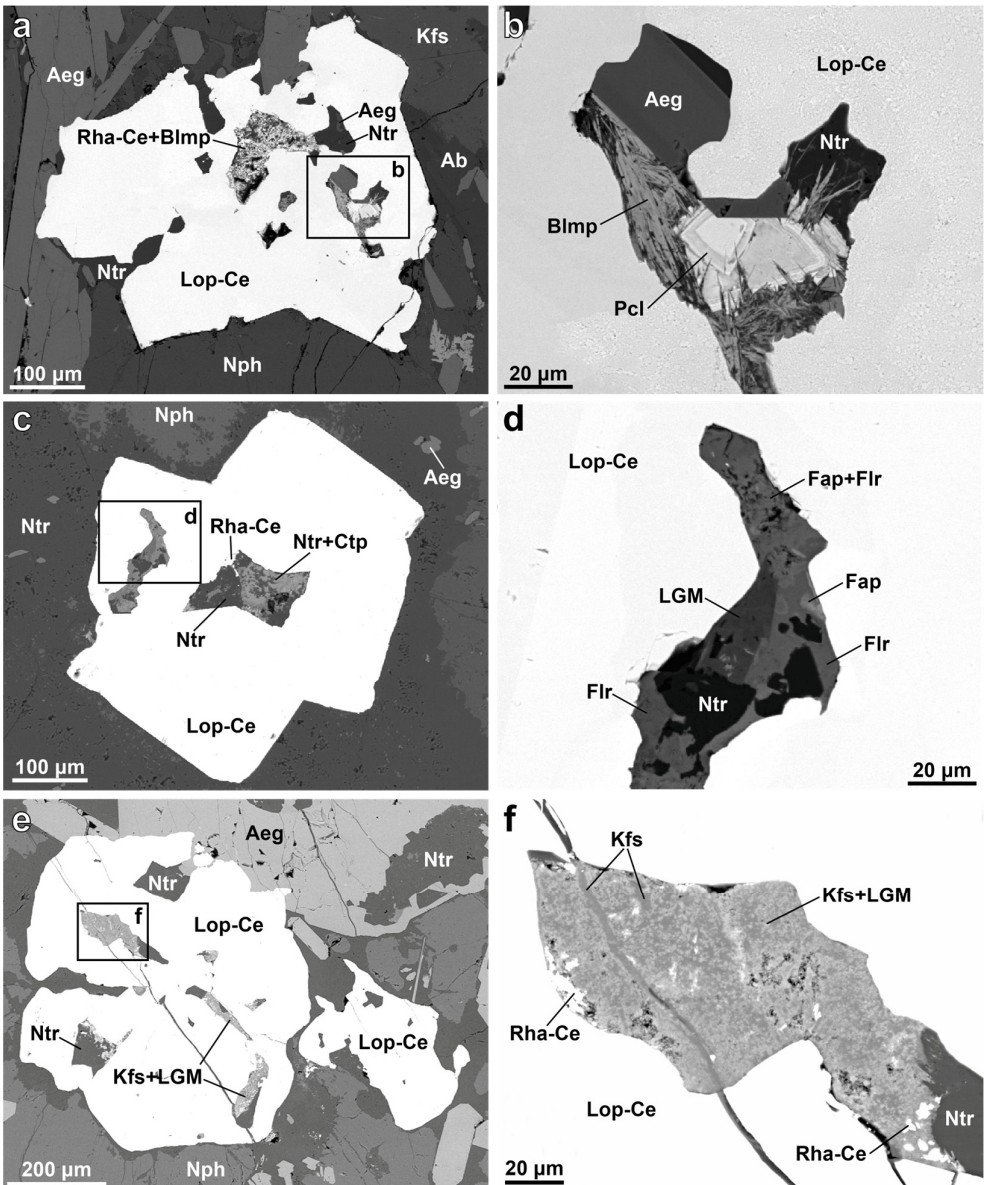

**Figure 5.** Back-scattered electron images of inclusions in loparite-(Ce) from the contact zones between rhythms: (**a**) general view of loparite-(Ce) grain with mineral inclusions and (**b**) detailed image of the inclusion (sample LV-316/1); (**c**) general view of loparite-(Ce) grain with mineral inclusions and (**d**) detailed image of the inclusion (sample LV-336/1); and (**e**) general view of loparite-(Ce) grain with mineral inclusions and (**f**) detailed image of the inclusion (sample LV-319D). See Table 2 for abbreviations.

A total of 21 mineral species and two groups of minerals (namely, pyrochlore- and labuntsovite-group minerals) were found in loparite-hosted inclusions (Table 4). This list includes both the minerals typical of rocks bearing loparite-(Ce) grains (i.e., groundmass minerals), such as aegirine, natrolite, and albite, as well as minerals that were not found in the rock outside of the loparite-(Ce) grains. The latter include lorenzenite, labuntsovite-group minerals, manganoneptunite, vinogradovite, etc., found in the studied samples exclusively inside loparite-(Ce) grains. It is important to note that the mineral associations in loparite-(Ce) inclusions from the contact zone between complexes and the contact zones between rhythms are different from each other (Table 4).

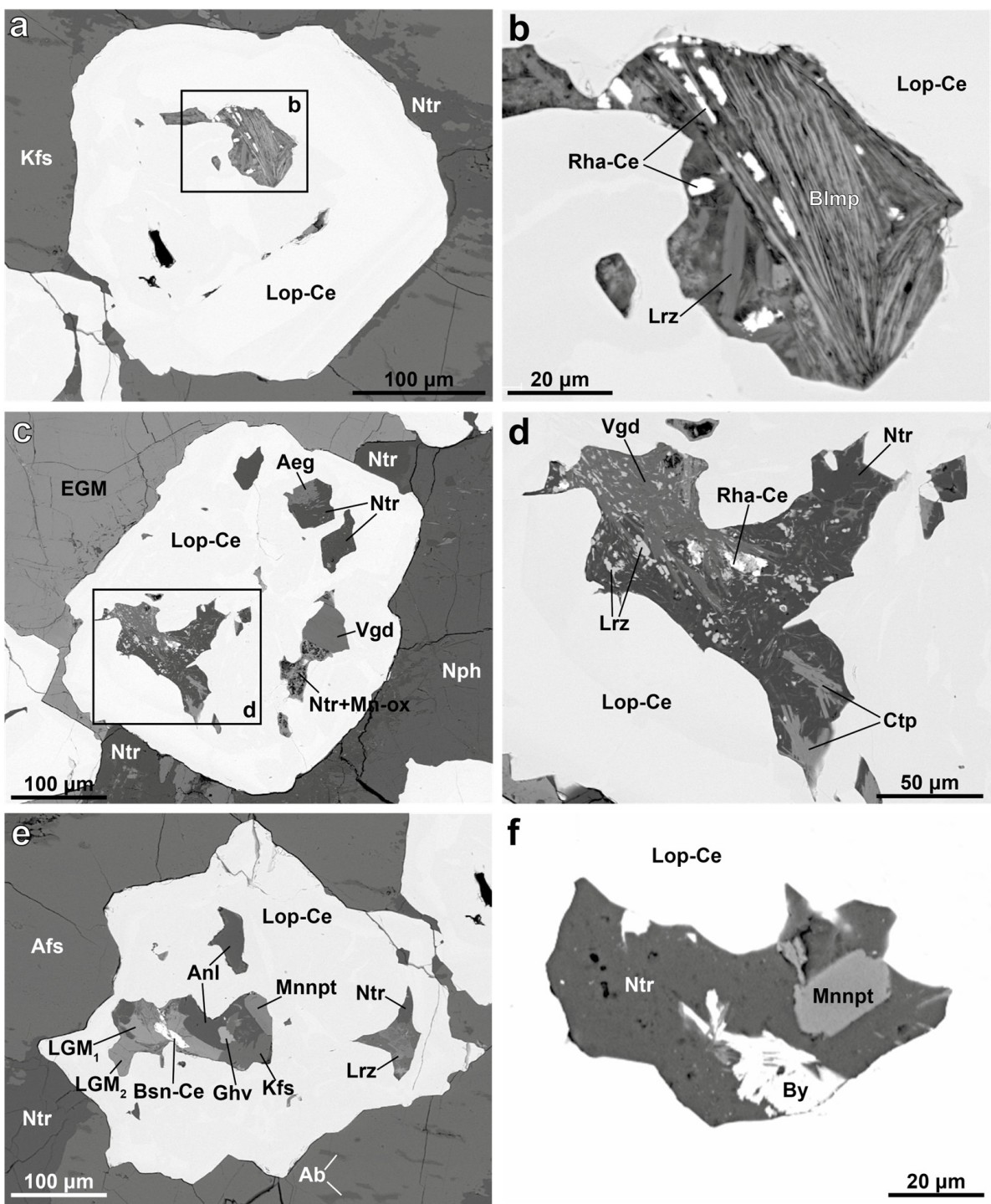

**Figure 6.** Back-scattered electron images of inclusions in loparite-(Ce) from the contact zone between complexes: (**a**) general view of loparite-(Ce) grain with mineral inclusions and (**b**) detailed image of the inclusion (sample LV-484/11); (**c**) general view of loparite grain with mineral inclusions and (**d**) detailed image of the inclusion (sample LV-484/11); (**e**) mineral associations in inclusions in loparite from sample LV-484/12. The chemical compositions of the labuntsovite-group minerals LGM$_1$ and LGM$_2$ are different (for details see Supplementary Materials, Table S7); and (**f**) inclusion in loparite-(Ce) from sample LV-484/12. See Table 2 for abbreviations.

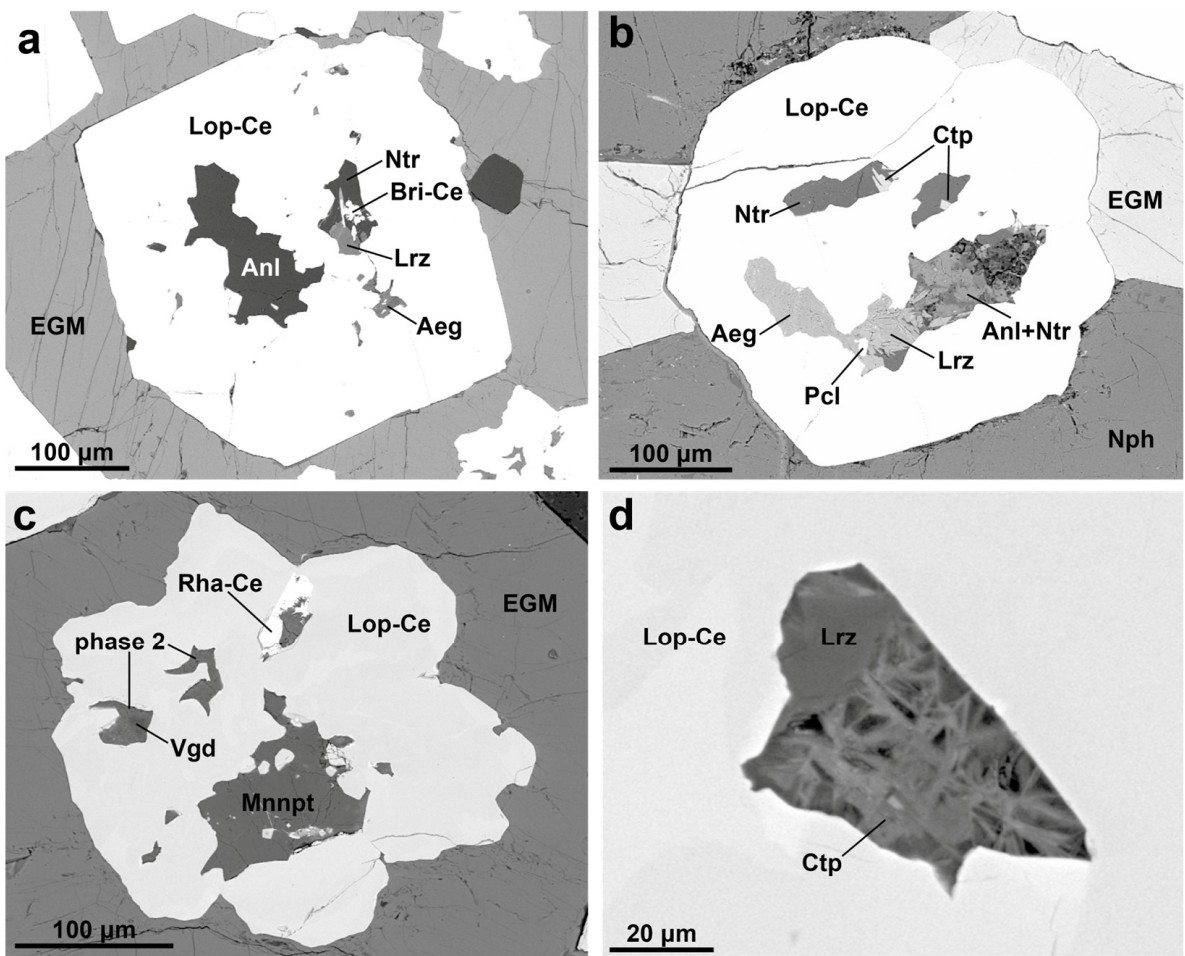

**Figure 7.** Back-scattered electron images of inclusions in loparite-(Ce) from the contact zone between complexes. (**a–d**) mineral associations in inclusions in loparite-(Ce) from sample LV-484/13. See Table 2 for abbreviations. phase 2 = undiagnosed Mn-Al-Si phase.

**Table 4.** List of minerals found in loparite-(Ce) inclusions.

| Mineral/Group of Minerals | Contact Zone between Complexes | | Contact Zones between Rhythms | |
|---|---|---|---|---|
| | In Inclusions | In Rock | In Inclusions | In Rock |
| natrolite | ● | ● | ● | ● |
| analcime | ● | ● | ● | ● |
| aegirine | ● | ● | ● | ● |
| magnesio-arfvedsonite | ● | ● | ● | ● |
| alkali feldspar | ● | ● | ● | ● |
| albite | ● | ● | ● | ● |
| fluorapatite | ● | ● | ● | ● |
| sodalite | ● | ● | ● | ● |
| rhabdophane-(Ce) | ● | ● | ● | ● |
| barytolamprophyllite | ● | ● | ● | ● |
| pyrochlore-group minerals | ● | ● | ● | ● |
| bastnäsite-(Ce) | ● | ● | ● | ● |

(minerals typical of rocks bearing loparite-(Ce) grains (groundmass minerals))

**Table 4.** *Cont.*

| Mineral/Group of Minerals | Contact Zone between Complexes | | Contact Zones between Rhythms | |
|---|---|---|---|---|
| | In Inclusions | In Rock | In Inclusions | In Rock |
| labuntsovite-group minerals | ● | | ● | |
| lorenzenite | ● | | ● | |
| catapleiite | ● | | ● | |
| vinogradovite | ● | | | |
| manganoneptunite | ● | | | |
| barylite | ● | | | |
| genthelvite | ● | | | |
| britholite-(Ce) | ● | | | |
| barite | ● | | | |
| fluorite | | | ● | |
| neptunite | | | ● | |

*Note: the left side of the table is labeled vertically:* minerals that were not found in the rock outside loparite-(Ce) grains

●—the mineral is present.

### 4.3. Minerals in Loparite-Hosted Inclusions

Natrolite is the most common mineral in inclusions, filling the spaces between other minerals (Figures 4a–e, 5, 6c–f, and 7a,b). Natrolite was diagnosed by chemical composition as well as Raman spectroscopy and X-ray diffraction data. Representative compositions of natrolite are shown in the Supplementary Materials, Table S1, and a typical Raman spectrum is shown in Figure 8a. Natrolite shows characteristic peaks at 443–445 $cm^{-1}$ and 533–536 $cm^{-1}$. In addition, there was another characteristic peak at 1039–1040 $cm^{-1}$. The strongest lines of the powder X-ray diffraction pattern of natrolite [d, Å (I, %)] were 2.85 (100), 5.89 (85), 2.87 (80), 4.35 (70), 6.55 (60), 3.16 (50), and 3.19 (45). Natrolite is characterized by a high $H_2O$ content (2.42–2.96 per formula unit), but X-ray data suggest that it was natrolite and not paranatrolite.

Analcime is found in intergrowths with natrolite, but this zeolite occurs as the only mineral in some inclusions (Figures 6e and 7a). Analcime was diagnosed by chemical composition as well as Raman spectroscopy and X-ray diffraction data. Representative chemical compositions of analcime are presented in the Supplementary Materials, Table S1. The strongest lines of the powder X-ray diffraction pattern of analcime [d, Å (I, %)] were 3.32 (100), 5.60 (60), and 2.93 (50). On the Raman spectra, analcime showed characteristic peaks at 298 $cm^{-1}$, 390 $cm^{-1}$, and 483 $cm^{-1}$.

Aegirine and magnesio-arfvedsonite form relatively large prismatic crystals (Figure 5a,b) as well as clusters of needle-like crystals and anhedral grains (Figure 4c). Representative compositions of aegirine and magnesio-arfvedsonite from inclusions in loparite-(Ce) are shown in the Supplementary Materials, Table S2. The chemical compositions of aegirine and magnesio-arfvedsonite in inclusions and the surrounding rock are the same. Titanium and Zr are typical impurities in aegirine (up to 0.10 Ti pfu and up to 0.03 Zr pfu), while titanium is a common impurity in amphibole (up to 0.11 Ti pfu).

Potassic feldspar forms relatively large (up to 150 μm across, Figure 4c) crystals or small grains in close intergrowth with labuntsovite-group minerals (Figure 5e,f). Albite is rare in inclusions and usually forms fine grains in natrolite. Like albite, sodalite is a rare mineral in inclusions; it usually forms intergrowths with natrolite. Representative compositions of potassic feldspar, albite, and sodalite from inclusions in loparite-(Ce) are shown in Supplementary Materials, Table S3.

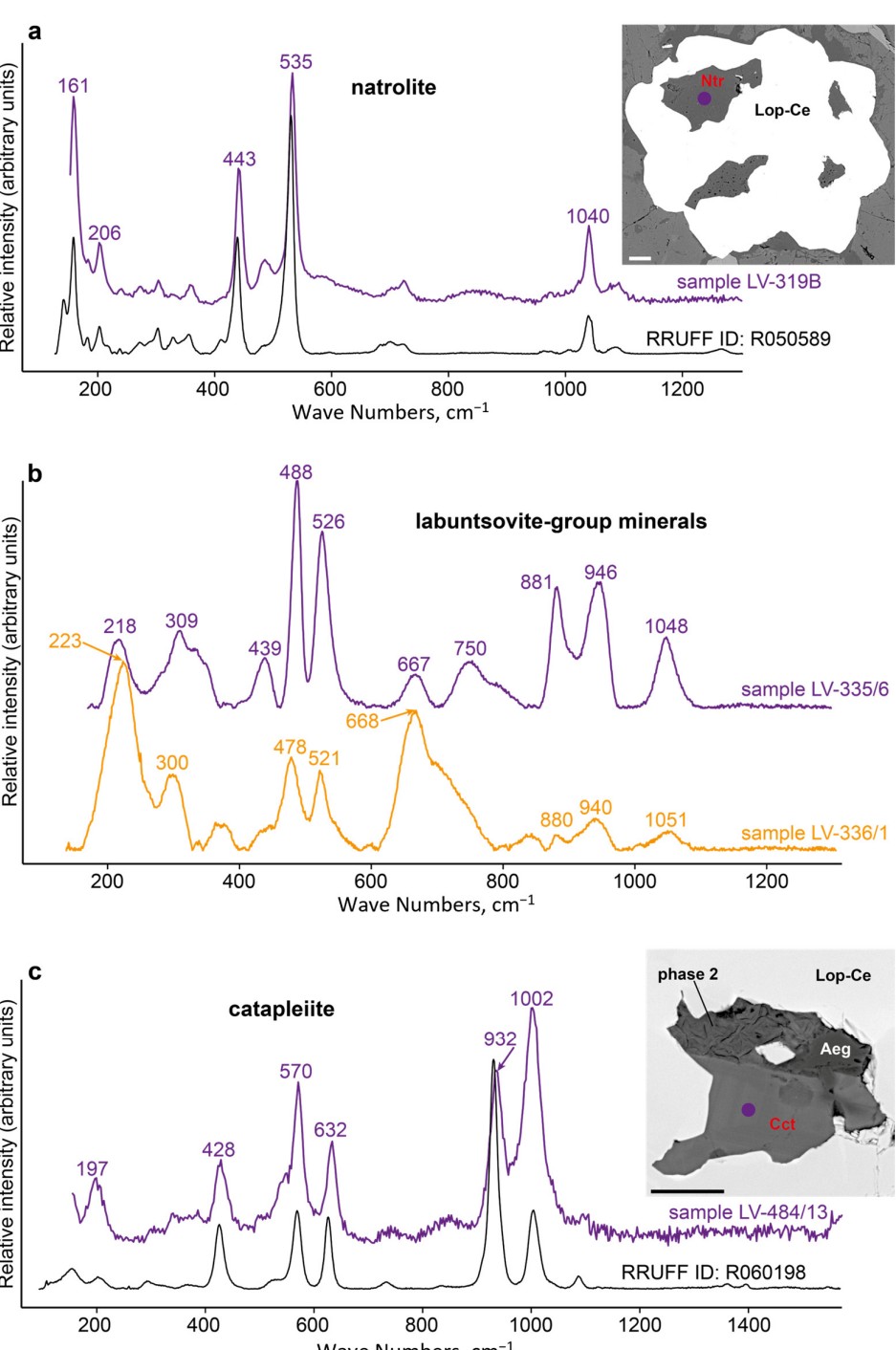

**Figure 8.** Representative Raman spectra of minerals found in loparite-hosted inclusions: (**a**) Raman spectrum of natrolite (left) and BSE image of the studied sample (right); (**b**) Raman spectra of labuntsovite-group minerals; and (**c**) Raman spectrum of catapleiite (left) and BSE image of the studied sample (right). The dots on the BSE images are analysis points. scale bars = 20 μm. See Table 2 for abbreviations.

Rhabdophane-(Ce) is widespread both in loparite-(Ce) inclusions and the surrounding rock (Figures 4, 5a,c, 6a,b,d, and 7c). This mineral usually forms small grains up to 20 μm across. Small (up to 20 μm across) rounded grains of fluorapatite are found in association with fluorite, natrolite, and labuntsovite-group minerals (Figure 5c,d). Representative compositions of rhabdophane-(Ce) and fluorapatite from loparite-(Ce) inclusions are shown in the Supplementary Materials, Table S4. Typical impurities in the composition of fluorapatite

include strontium (up to 0.50 Sr pfu) and rare earth elements (up to 0.10 Ce pfu and up to 0.05 La pfu).

In rocks, barytolamprophyllite forms rims around lamprophyllite crystals, while lamprophyllite was not found in loparite-(Ce) inclusions, and barytolamprophyllite appears as fine needle-like crystals (Figures 5a,b and 6a,b). Representative compositions of barytolamprophyllite from loparite-(Ce) inclusions are shown in the Supplementary Materials, Table S5. Lorenzenite was found only in loparite-hosted inclusions. Here, it is very widespread and forms small crystals (Figures 6b and 7b) and anhedral grains up to 50 μm across. The chemical composition of lorenzenite corresponds to the ideal formula, only minor impurities of niobium (up to 0.10 Nb pfu) and iron (up to 0.05 $Fe^{3+}$ pfu) are observed (Supplementary Materials, Table S5).

Pyrochlore-group minerals in inclusions usually form small (10 μm on average) rounded grains; less often, zoned crystals up to 30 microns across can be found (Figure 5a,b). Due to the small sizes of the zoned crystals, we measured only the chemical composition of the central parts. Representative compositions of pyrochlore-group minerals from loparite-(Ce) inclusions are shown in the Supplementary Materials, Table S6.

Labuntsovite-group minerals were found only in loparite-(Ce) inclusions. These minerals are very widespread here and are found in a wide variety of associations (Figures 4a,b,d–f and 5c–f). The chemical composition of labuntsovite-group minerals also varies widely (Supplementary Materials, Table S7). The same inclusion may contain labuntsovites of different chemical compositions (Figure 6e and Supplementary Materials, Table S7). Representative Raman spectra of labuntsovite-group minerals are shown in Figure 8b. LGM showed characteristic peaks at 218–223, 300–309, 478–488, 521–526, 667–668, 880–881, and 940–946 $cm^{-1}$.

Catapleiite is a typical mineral in inclusions (Figures 4a,b, 6c,d and 7b,d). It usually forms small laths or lamellar crystals (up to 30 μm across). This mineral was diagnosed by its chemical composition (Supplementary Materials, Table S8) and Raman spectroscopy data (Figure 8c). The calcium content in catapleiite does not exceed 0.10 apfu. Typical impurities are titanium (up to 0.08 Ti pfu) and potassium (up to 0.07 K pfu). All samples of catapleiite have a very low sodium content (0.95–1.33 Na pfu). The fact that the leaching of Na from many zeolite-like rare-element minerals is a process readily occurring at the late-hydrothermal evolution stages of alkaline massifs is supported by abundant evidence [29]. With sodium leaching, a deficiency of positive charge can be compensated in several ways. For example, sodium leaching can be accompanied by the appearance of a vacancy with the simultaneous replacement of oxygen by an OH group: $Na^+ + O^{2-} \rightarrow \square + OH^-$. Another way is the replacement of sodium with a water molecule with the simultaneous replacement of oxygen with an OH group: $Na^+ + O^{2-} \rightarrow H_2O + OH^-$. In catapleiite, the decrease in sodium content is probably also associated with leaching.

Vinogradovite was found only in loparite-(Ce) inclusions from the contact zone between the complexes (Table 4). It forms acicular (Figure 6c,d), flattened prismatic crystals up to 70 μm long or anhedral grains (Figure 6c) up to 50 μm across. Representative compositions of vinogradovite are presented in the Supplementary Materials, Table S8.

Minerals of the neptunite-manganoneptunite series were found in the studied samples only in loparite-(Ce) inclusions (Figures 6e and 7c). Neptunite-manganoneptunite was diagnosed by Raman spectroscopy (Figure 9a) and chemical composition (Supplementary Materials, Table S9). Minerals show characteristic peaks at 306–309, 330–333, 686–688, and 823–824 $cm^{-1}$. The ratio of ferrous iron and manganese varies widely. At the same time, neptunite-manganoneptunite with different $Fe^{2+}$/Mn ratios were found inside neighboring loparite-(Ce) grains. A typical impurity in the composition of these minerals is zinc (up to 0.68 Zn pfu).

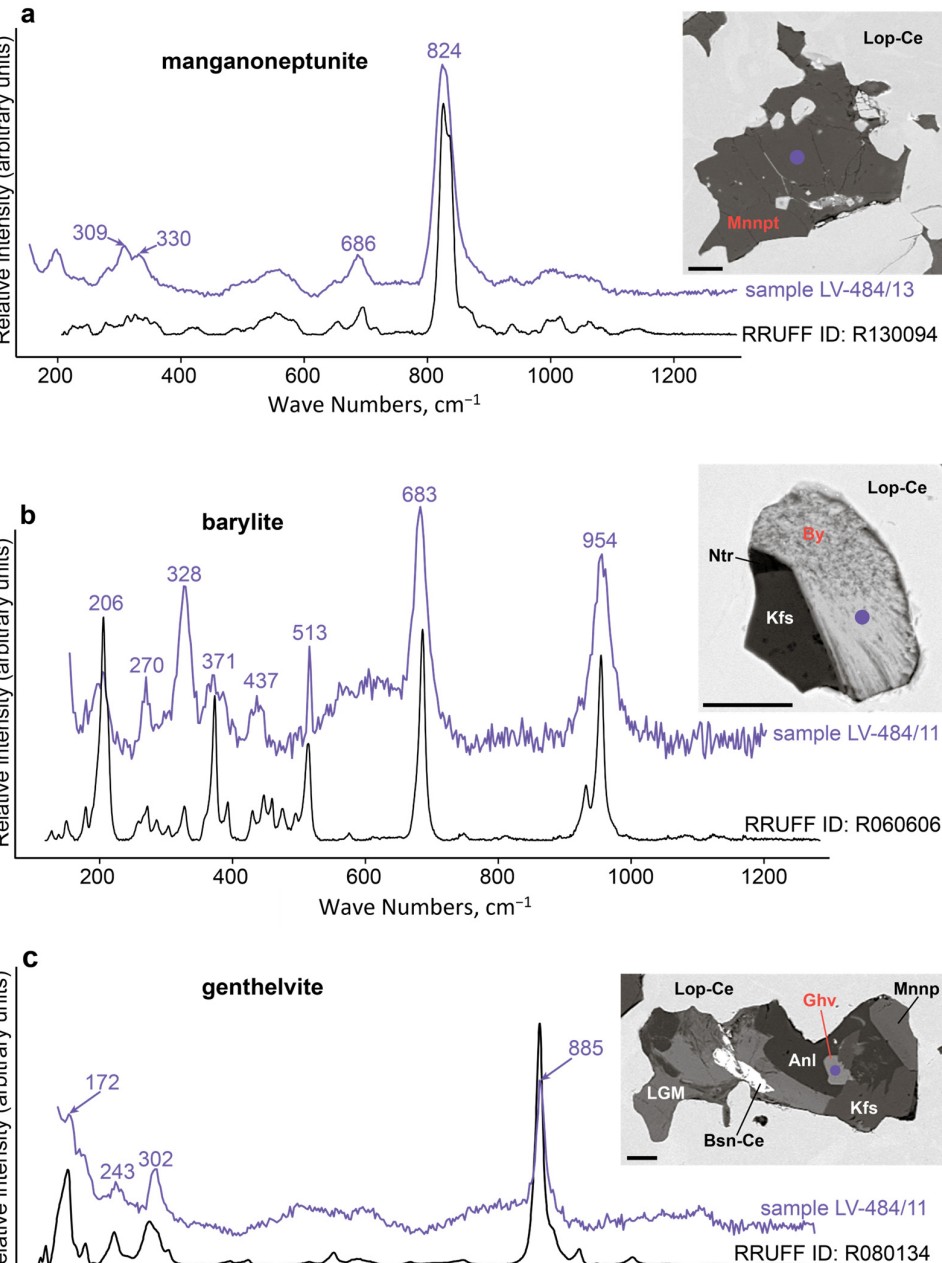

**Figure 9.** Representative Raman spectra of minerals found in loparite-hosted inclusions: (**a**) Raman spectrum of manganoneptunite (left) and BSE image of the studied sample (right); (**b**) Raman spectra of barylite (left) and BSE image of the studied sample (right); and (**c**) Raman spectrum of genthelvite (left) and BSE image of the studied sample (right). The dots on the BSE images are analysis points. scale bars = 20 μm. See Table 2 for abbreviations.

Barylite was found only in loparite-(Ce) inclusions from the contact zone between the complexes (Table 4). This mineral was identified by both Raman spectroscopy data (Figure 9b) and chemical composition (Supplementary Materials, Table S10). In the Raman spectra, barylite shows characteristic peaks at 206, 270, 328, 371, 437, 513, 683, and 954 cm$^{-1}$. The mineral contains minor impurities of potassium (up to 0.05 K pfu), iron (up to 0.02 Fe$^{3+}$ pfu), and aluminum (up to 0.03 Al pfu).

Genthelvite is a rare mineral in loparite-(Ce) inclusions. It was found in only one sample (Figure 6e) and diagnosed by Raman spectroscopy (Figure 9c). A very intensive

band at 170 cm$^{-1}$ should be assigned to Be–O vibrations [30]. According to [31,32], wt. % Be in genthelvite compositions may be estimated assuming that Si + Be = 6. The chemical composition of genthelvite is SiO$_2$ 31.10; TiO$_2$ 0.13; Ce$_2$O$_3$ 0.18; ZnO 46.45; MnO 0.52; BeO (calc.) 12.95; S 4.78; O = S 2.38. The total of 93.73 corresponds to the formula Be$_3$Zn$_{3.31}$Mn$_{0.04}$Ti$_{0.01}$Ce$_{0.01}$(SiO$_4$)$_3$S$_{0.86}$.

Bastnäsite-(Ce) is a widespread mineral found both in loparite-(Ce) inclusions and the surrounding rock. This mineral forms tiny (5 μm on average) rounded or irregularly shaped grains (Figure 6e). The chemical composition of bastnäsite-(Ce) from sample LV-484/11 (wt. %) was as follows: TiO$_2$ 0.56; La$_2$O$_3$ 19.21; Ce$_2$O$_3$ 39.94; Pr$_2$O$_3$ 3.21; Nd$_2$O$_3$ 9.00; SrO 0.78; F 6.07; O = F 2.55. The total of 76.22 corresponds to the formula (Ce$_{0.47}$La$_{0.23}$Nd$_{0.10}$Pr$_{0.04}$Sr$_{0.01}$Ti$_{0.01}$)$_{0.86}$(CO$_3$)F$_{0.62}$. This formula is based on (CO$_3$)$^{2-}$ = 1.

Britholite-(Ce) (Figure 7a) is a rare mineral in loparite-(Ce) inclusions. The chemical composition of this mineral from sample LV-484/13 (wt. %) was as follows: P$_2$O$_5$ 9.05; SiO$_2$ 15.59; ThO$_2$ 2.63; La$_2$O$_3$ 14.71; Ce$_2$O$_3$ 30.10; Pr$_2$O$_3$ 2.70; Nd$_2$O$_3$ 6.91; Sm$_2$O$_3$ 0.83; CaO 10.03; SrO 3.75; Na$_2$O 2.19; F 1.23; and O = F 0.52; the total of 99.20 corresponds to the formula (Ce$_{1.42}$Ca$_{1.39}$La$_{0.70}$Na$_{0.54}$Nd$_{0.32}$Sr$_{0.28}$Pr$_{0.13}$Th$_{0.08}$Sm$_{0.04}$)$_{4.90}$(Si$_{2.01}$P$_{0.99}$)$_{3.00}$O$_{12}$(F$_{0.50}$OH$_{0.50}$)$_{1.00}$. This formula is based on Si + P = 3 apfu.

## 5. Discussion

The habits of crystals growing in liquids are controlled by the degree of supersaturation, i.e., the difference between the concentration of a chemical component at equilibrium and the actual concentration in the liquid at a specified temperature and pressure [33–37]. The reason is that when supersaturation changes, the mechanism of crystal growth changes too. If the degree of supersaturation is very low, then the crystals grow by screw dislocations. In this case, slightly convex faces with dislocation hillocks on an almost flat surface are formed. As the degree of supersaturation increases, the main growth mechanism becomes two-dimensional nucleation on the crystal faces. As a result, the crystal has flat faces with roughness where two-dimensional islands are located. At a very high degree of supersaturation, the edges and vertices of the crystal grow faster than the faces [37,38]. The resulting crystals are characterized by fully developed crystal edges with hollow interiors. Figure 10a shows the relationships between growth rate, surface growth mechanism, and supersaturation.

The appearance of inclusion-bearing loparite-(Ce) can be explained by assuming the skeletal or dendritic growth of its crystals. In Figure 10b, we correlated the typical loparite-(Ce) morphology from the studied samples with the mechanism of surface growth and the degree of supersaturation. The identical chemistry of inclusion-free loparite-(Ce) from the Kedykvyrpakhk mine and inclusion-bearing loparite-(Ce) from the Karnasurt mine (Figure 3d and [11]) indicates that they crystallized from the analogous parental liquid but under different degrees of supersaturation. Loparite-(Ce) crystals from the Kedykvyrpakhk mine crystallized at a relatively low degree of supersaturation, while loparite-(Ce) crystals from the Karnasurt mine grew at a higher degree of supersaturation. Loparite-(Ce) from 3–5 sampling areas, such as those from the Karnasurt mine, crystallized at a high degree of supersaturation in the form of skeletal crystals (Figures 4a,b,d–f and 5–7).

Thus, we believe that loparite-(Ce) saturated with inclusions initially crystallized in the form of skeletal crystals. This made it possible to capture both co-crystallizing minerals and droplets of a melt (or solution). Such droplets subsequently crystallized within the loparite-(Ce), leading to the formation of polymineralic inclusions. A similar mechanism for the formation of polymineralic inclusions has been established for chromite from the Jacurici Complex, Brazil [39] and from the Troodos ophiolite [40].

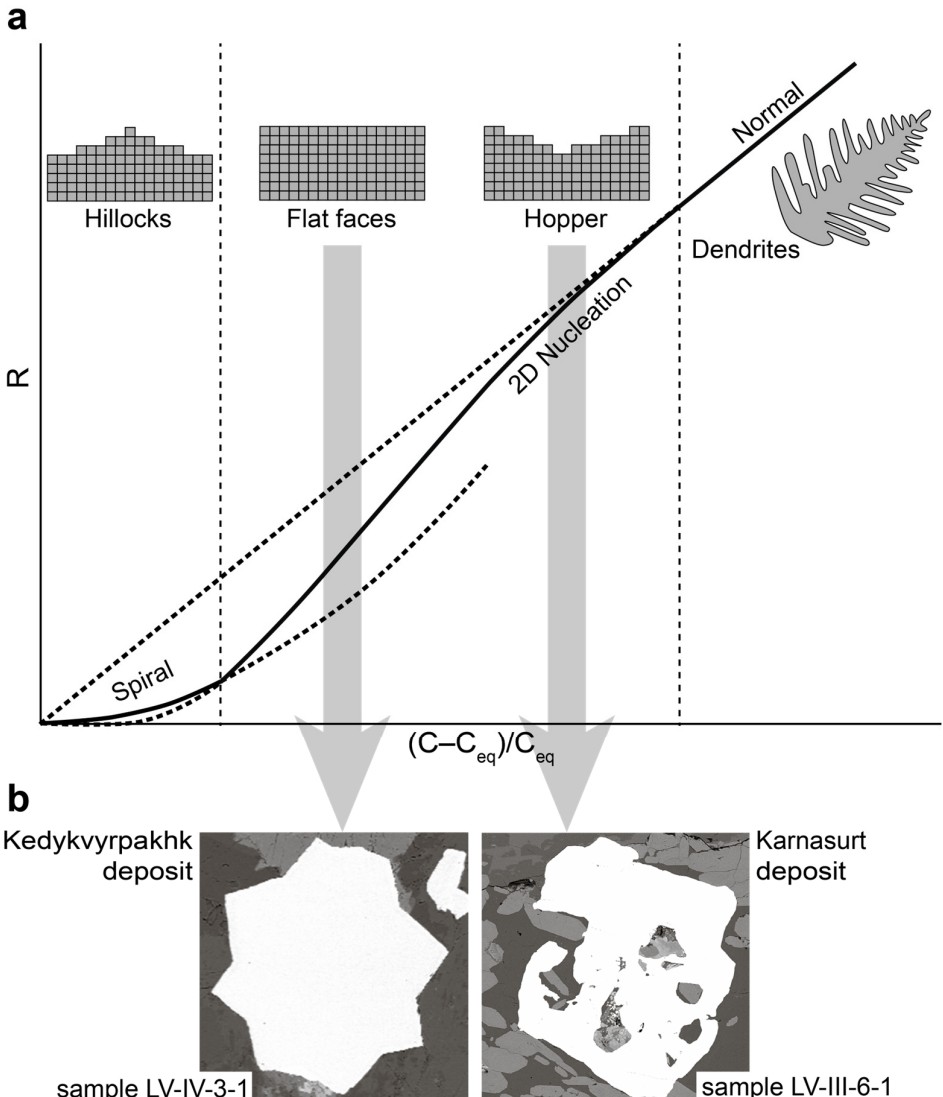

**Figure 10.** Growth rate, surface growth mechanism, and loparite-(Ce) morphology as a function of supersaturation. (**a**) The relationship between growth rate (R), surface growth mechanism, and supersaturation (after [37,38] with modifications). C—actual concentration of a chemical component in the liquid; $C_{eq}$—concentration at equilibrium. Vertical dashed lines separate the spiral growth, two-dimensional (2D) nucleation, and normal growth regimes. The upper part of the figure shows the crystal surface morphologies for the spiral and 2D nucleation mechanisms, as well as the morphology of dendritic crystals. (**b**) The changes in the loparite-(Ce) morphology from the studied samples depend on the degree of supersaturation. BSE mages.

A total of 21 mineral species and two groups of minerals (pyrochlore- and labuntsovite-group minerals) were found in loparite-(Ce) inclusions (Table 4). Minerals found in loparite-(Ce) inclusions can be subdivided into the following groups: (1) minerals typical of rocks bearing loparite-(Ce) grains (groundmass minerals) such as aegirine, magnesio-arfvedsonite, potassic feldspar, albite, fluorapatite, etc.; and (2) minerals that were not found in the rock outside of loparite-(Ce) grains. The latter include lorenzenite, labuntsovite-group minerals, neptunite-manganoneptunite, vinogradovite, catapleiite, fluorite, britholite-(Ce), barylite, genthelvite, and barite, found in the studied samples exclusively inside loparite-(Ce) grains.

The minerals of the second group are typical hydrothermal minerals. In fact, the labuntsovite-group minerals are always of late-hydrothermal origin. These minerals are typical of various alkaline pegmatites and hydrothermal veins, as well as carbonatites [41–46].

The formation of labuntsovite-group minerals from alkaline hydrothermal solutions is stimulated by the high activity of silica [47].

　　Lorenzenite, in the Lovozero massif, is a rare mineral in rocks, but it is widespread in pegmatites and rocks in the exocontact zone of the massif [48,49]. Catapleiite in agpaitic rocks and related pegmatites replaces eudialyte-group minerals and other zirconosilicates at late-hydrothermal stages [48,50]. Minerals of the neptunite-manganoneptunite series are low-temperature hydrothermal minerals [51]. In pegmatites of the Lovozero massif, manganoneptunite is associated with ussingite, natrolite, and analcime, and forms finely faceted crystals growing on the walls of cavities [48,52]. Vinogradovite is a typical hydrothermal mineral that crystallizes in the cavities of pegmatites and hydrothermal veins; it often replaces other titanosilicates, especially lorenzenite [48,49]. Natural occurrences of genthelvite are mainly restricted to pegmatites associated with highly fractionated alkaline to peralkaline granites and syenites [53]. In the Lovozero massif, genthelvite was found in pegmatites (Selsurt Mt., northern part of the massif) in association with orthoclase, ilmenite, and zircon [54]. Barylite is found mainly in skarn deposits associated with granitoids and hydrothermal veins associated with alkaline massifs, especially in veins located in the exo- and endocontact zones of these massifs [55].

　　It is important to note that nepheline is absent from the list of minerals (Table 4) found in loparite-(Ce) inclusions. Moreover, nepheline was not found in bay-like cavities, which could be interpreted as 'incompletely captured' inclusions (Figures 2d,f, 3b,c and 5a,e). Meanwhile, nepheline is a rock-forming mineral in all the studied samples (for example, Figures 2a–c,e and 3a), and in some samples, the content of this mineral reaches 60 vol. %. It can be assumed that during the formation of loparite-(Ce), nepheline has not yet crystallized, but a typical mineral of loparite-hosted inclusions is aegirine (Figures 4c and 5b), which crystallized later than nepheline [18,20,56]. Therefore, it cannot be assumed that loparite-(Ce) crystallized before nepheline.

　　In all lithologies of the Lovozero massif, natrolite is a typical and widespread product of nepheline alteration [48,49]. At the same time, both in inclusions and in bay-like cavities, natrolite is constantly present (for example Figures 2d,f and 8a). Skeletal loparite-(Ce) crystals are usually surrounded by a natrolite rim and located in areas where nepheline is intensively pseudomorphized by natrolite and other minerals (Figures 2b,c,e and 5c,e). Additionally, in Figure 11, we have shown such relationships between loparite-(Ce) and natrolite by changing the contrast of the BSE image.

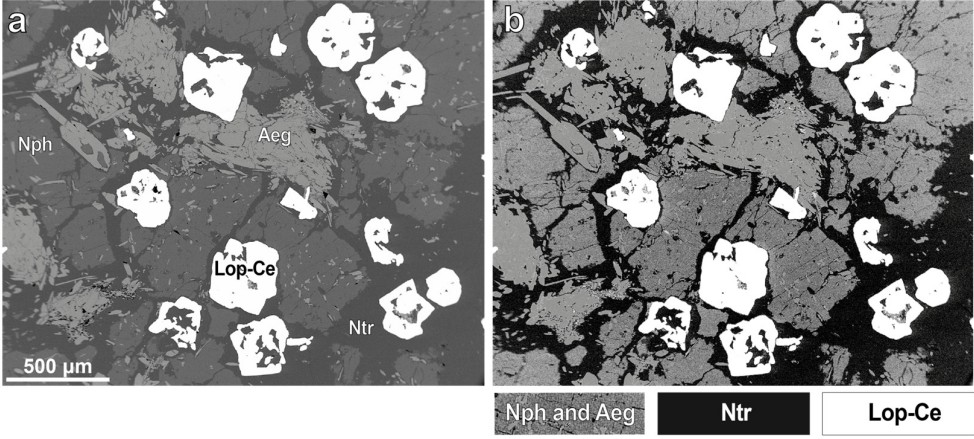

**Figure 11.** Skeletal loparite-(Ce) crystals are surrounded by a natrolite rim or located within the natrolite mass (bottom right). (**a**) Back-scattered electron image of sample LV-III-6-1; and (**b**) back-scattered electron image of sample LV-III-6-1 with changed contrast. See Table 2 for abbreviations.

　　These facts indicate that skeletal loparite-(Ce) crystallized almost simultaneously with natrolite in late-magmatic/hydrothermal stages [11,13]. Such loparite-(Ce) captured co-crystallizing (natrolite, albite, and secondary feldspar) and possibly previously crystallized

(aegirine and magnesio-arfvedsonite) minerals as well as small droplets of the hydrothermal solution. These droplets subsequently crystallized within inclusions to form hydrothermal minerals such as labuntsovite-group minerals, lorenzenite, neptunite-manganoneptunite, vinogradovite, catapleiite, etc.

Why did the portions of hydrothermal solution that were trapped in loparite-(Ce) produce different Ti-, Zr-, and REE-bearing mineral species that the rest of the same solution outside did not? We assume that the crystallization of loparite-(Ce) significantly depleted the hydrothermal solution of Ti, Nb, and Ta, as well as rare earth elements. In addition, the water content in the solution increased since it was not included in the composition of the crystallizing loparite-(Ce). The inclusions were formed as a result of the isolation of portions of the hydrothermal solution during the crystallization of loparite-(Ce), when this solution was still enriched in Ti, Nb, Ta, and rare earth elements.

It is also important to note the difference in the mineral associations of loparite-hosted inclusions from the contact zones between rhythms and the contact zone between complexes (Table 4). The Lovozero massif is widely known for its pegmatites and hydrothermal veins with uniquely diverse mineralogy. Pegmatites are located mainly in two types of settings: (1) at the contacts between rhythms of the layered complex and (2) at the contacts between the poikilitic complex and surrounding alkaline rocks [48,49,57]. Various Li- and Be-bearing minerals are typical of pegmatites spatially associated with rocks of the poikilitic complex. Minerals such as bertrandite, beryllite, chkalovite, epididymite, manganoneptunite and others were found here. At the same time, lithium and beryllium minerals, namely barylite, genthelvite, and manganoneptunite, were found in loparite-(Ce) from the contact zone between the poikilitic and the layered complexes. Thus, the mineral associations in loparite-hosted inclusions mimic the geochemical features of hydrothermal associations.

## 6. Conclusions

A total of 21 mineral species and two groups of minerals (pyrochlore- and labuntsovite-group minerals) were found in polymineralic loparite-(Ce) inclusions from alkaline rocks of the Lovozero massif (Russia). Minerals in loparite-hosted inclusions can be divided into two groups: (1) minerals typical of rocks bearing loparite-(Ce) grains such as aegirine, magnesio-arfvedsonite, potassic feldspar, albite, fluorapatite, etc.; and (2) minerals that were not found in the rock outside of loparite-(Ce) grains. The latter include lorenzenite, labuntsovite-group minerals, minerals of the neptunite-manganoneptunite series, vinogradovite, catapleiite, fluorite, britholite-(Ce), barylite, genthelvite, and barite, found in the studied samples exclusively inside loparite-(Ce) crystals. The minerals of the second group are typical hydrothermal minerals. Loparite-(Ce) in the rocks of the Lovozero massif crystallized in the late-magmatic/hydrothermal stage, and the morphology of its crystals depended on the degree of supersaturation. At a comparatively high degree of supersaturation, loparite-(Ce) grew in the form of skeletal crystals, capturing both co-crystallizing (natrolite, albite, and secondary feldspar) and possibly previously crystallized (aegirine and magnesio-arfvedsonite) minerals as well as small droplets of the hydrothermal solution. Such droplets subsequently crystallized within the loparite-(Ce), resulting in the formation of a hydrothermal mineral association.

**Supplementary Materials:** The following supporting information can be downloaded at: https://www.mdpi.com/article/10.3390/min13060715/s1, Table S1: Representative compositions of zeolites; Table S2: Representative compositions of aegirine and magnesio-arfvedsonite; Table S3: Representative compositions of K-feldspar, albite, and sodalite; Table S4: Representative compositions of rhabdophane-(Ce) and fluorapatite; Table S5: Representative compositions of barytolamprophyllite and lorenzenite; Table S6: Representative compositions of pyrochlore-group minerals; Table S7: Representative compositions of labuntsovite-group minerals; Table S8: Representative compositions of catapleiite and vinogradovite; Table S9: Representative compositions of neptunite and manganoneptunite; Table S10: Representative compositions of barylite.

**Author Contributions:** Conceptualization, J.A.M. and Y.A.P.; methodology, J.A.M. and Y.A.P.; software, A.A.K.; investigation, Y.A.P., E.A.S. and A.A.K.; data curation, J.A.M. and E.A.S.; writing—original draft preparation, J.A.M.; writing—review and editing, J.A.M. and E.A.S.; visualization, J.A.M. and A.A.K. All authors have read and agreed to the published version of the manuscript.

**Funding:** This research was funded by the Russian Science Foundation, project no. 21-47-09010.

**Data Availability Statement:** Not applicable.

**Acknowledgments:** We are grateful to the reviewers who helped us improve the presentation of our results.

**Conflicts of Interest:** The authors declare no conflict of interest.

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
