# Peer review of "Polymineralic Inclusions in Loparite-(Ce) from the Lovozero Alkaline Massif (Kola Peninsula, Russia): Hydrothermal Association in Miniature"

_minerals, doi:10.3390/min13060715_

Round 1
Reviewer 1 Report
Comments to the manuscript (MS).
1. The conclusions must be extended with addition of some important observations from the MS discussion.
2. Line 358-360 (p.18): The hydration schemes are not balanced stoichiometrically. The schemes must be corrected or explained in necessary detail.
3. Lines 147-149 (p.5): The text is a repetition of note#1 to Table 2.
Author Response
We are very grateful to the Reviewer for careful revision of our manuscript and important comments.
Point 1. The conclusions must be extended with addition of some important observations from the MS discussion.
Response 1. The conclusions were extended with addition of observations from discussion (lines 523-532 and 534-537). In particular, a brief mineralogical description of loparite-hosted inclusions has been added.
Point 2. Line 358-360 (p.18): The hydration schemes are not balanced stoichiometrically. The schemes must be corrected or explained in necessary detail.
Response 2. A detailed explanation of hydration schemes has been added to the text (lines 354-361).
Point 3. Lines 147-149 (p.5): The text is a repetition of note#1 to Table 2.
Response 3. Note#1 removed.
Reviewer 2 Report
It was pleasure for me to read this really very interesting paper. I recommend publishing it in its present form, which is unique case in my over 40 years long experience as a reviewer.
I have only one question, that would be fine to be discussed in few additional sentences:
Why the portions of hydrothermal solution, trapped in loparite, produced some mineral species, that the rest of the same solution outside didn't? In other words, what are the specific condidtions inside loparite crystals, that promote the formation of these phases? Providing elemental support from the loparite matrix? Providing epitaxial benefit for nucleation? Or ... ?
Author Response
We are very grateful to the Reviewer for the high appraisal of our work!
Why the portions of hydrothermal solution, trapped in loparite-(Ce), produced different Ti-, Zr-, REE-bearing mineral species, that the rest of the same solution outside didn't? We have added a few sentences to discuss this (lines 501-508).
We assume that the crystallization of loparite-(Се) significantly depleted the hydrothermal solution in Ti, Nb, Ta, as well as rare earth elements. In addition, the water content in the solution increased, since it was not included in the composition of the crystallizing loparite-(Се). The inclusions were formed as a result of the isolation of portions of the hydrothermal solution during the crystallization of loparite-(Се), when this solution was still enriched in Ti, Nb, Ta, and rare earth elements.